# Super-Resolution Reconstruction of Terahertz Images Based on Residual Generative Adversarial Network with Enhanced Attention

**DOI:** 10.3390/e25030440

**Published:** 2023-03-02

**Authors:** Zhongwei Hou, Xingzeng Cha, Hongyu An, Aiyang Zhang, Dakun Lai

**Affiliations:** School of Electronic Science and Engineering, University of Electronic Science and Technology of China, Chengdu 611731, China

**Keywords:** terahertz image, image super-resolution, generative adversarial network, attention mechanism

## Abstract

Terahertz (THz) waves are widely used in the field of non-destructive testing (NDT). However, terahertz images have issues with limited spatial resolution and fuzzy features because of the constraints of the imaging equipment and imaging algorithms. To solve these problems, we propose a residual generative adversarial network based on enhanced attention (EA), which aims to pay more attention to the reconstruction of textures and details while not influencing the image outlines. Our method successfully recovers detailed texture information from low-resolution images, as demonstrated by experiments on the benchmark datasets Set5 and Set14. To use the network to improve the resolution of terahertz images, we create an image degradation algorithm and a database of terahertz degradation images. Finally, the real reconstruction of terahertz images confirms the effectiveness of our method.

## 1. Introduction

Terahertz refers to electromagnetic waves with frequencies between 100 GHz and 10,000 GHz. The waveband is between microwaves and infrared light. It has some unique properties such as transience, low energy, and penetrability. Therefore, terahertz imaging [1] can be applied in fields such as biomedical diagnosis [2,3], non-destructive testing [4], industrial safety testing [5,6], etc.

A crucial technique for terahertz imaging is terahertz tomography [7]. Terahertz tomography makes use of the penetrability of terahertz waves through dielectrics to capture the changes in amplitude and phase of the electromagnetic waves as they pass through the object. Then, tomographic reconstruction algorithms are used to take these changes and figure out how the object is made inside. 

However, terahertz tomography images have certain issues that result in poor resolution. Firstly, the reconstructed image is distorted due to the diffraction and scattering of the terahertz waves inside the object. Secondly, because the data acquisition process is discrete, the data in the reconstructed image are also discrete, and the missing data lead to ray-like stripe artifacts in the image.

In addition, due to the strong energy in the center of the THz Gaussian beam and weak energy in the periphery, and the fact that the beam has a certain width, the edges of the reconstructed images are presented as bands with significant widths rather than sharp lines. In other words, the picture quality is poor, and the edge’s contour is hazy. Therefore, improving the terahertz image’s quality after reconstruction is a crucial challenge.

There are mainly two ways to improve the resolution of terahertz images. One is to improve the imaging system, and the other is to adopt a super-resolution reconstruction method. At present, some research teams focus on improving the terahertz imaging system to improve the spatial resolution [8,9,10], while other teams propose to obtain a better image effect via a super-resolution algorithm. Li et al. [11] used the Lucy–Richardson algorithm to improve the resolution of coherent terahertz imaging systems. Ding et al. [12] applied the Lucy–Richardson algorithm to a reflective terahertz imaging system.

In recent years, Dong et al. [13] have proposed a super-resolution reconstruction method based on a convolutional neural network (CNN). They applied a three-layer super-resolution network and achieved state-of-the-art results in a super-resolution reconstruction at that time. Since then, many deeper and better-performing convolutional network models for super-resolution reconstruction have been proposed [14,15,16,17]. Meanwhile, there were research teams that applied convolutional super-resolution networks to terahertz image reconstruction [18,19,20]. However, these methods constructed the network model by increasing the number of convolutional layers, causing the parameters to expand exponentially and they did not introduce specific structures to enhance the reconstruction of image details. Deeper networks can extract more information for picture reconstruction, but network overfitting makes training difficult. After that, Fan et al. [21] proposed a lightweight SRCNN method to improve the image quality and to meet the demand of small datasets. In 2021, Su et al. [22] suggested a novel subspace-and-attention-guided restoration network (SARNet), which adopted attention guidance to fuse spatio-spectral features of amplitude and phase. Yang et al. [23] introduced the remaining channel mechanism and the residual channel attention mechanism to restore the high-frequency information. 

This article proposes a super-resolution reconstruction model for terahertz images based on the residual generative adversarial network with EA. The model has a multibranch residual block convolutional structure that obtains feature information from each layer during the feature extraction process and conducts feature fusion. Additionally, an enhanced attention mechanism that combines both spatial and channel attention has been added to the residual block. In addition to extracting information from the channels, it also incorporates direction-aware and position-sensitive information, forcing the network to focus more on texture and picture details while maintaining the integrity of the image’s shape and lowering network parameters. 

The main contributions of this paper are summarized as follows:We design a super-resolution generative adversarial network with attention and residuals that are suitable for multiple super-resolution tasks.We employ an enhanced attention mechanism and make the network pay more attention to the reconstruction of image details and texture information.We use the cosine annealing algorithm to improve the network training process, speed up the training process, and effectively improve the network’s performance.We build a terahertz degradation model and image database, and apply the network to terahertz tomography image super-resolution reconstruction.

## 2. Related Work

### 2.1. Deep CNN Super-Resolution Based on Residual Block

Due to the development of integrated circuits and the increasing GPU computing power, deep learning has been gradually applied in every field. Dong et al. [13] applied the deep learning method to SR and acquired a far more effective image SR compared with the traditional methods. In 2016, the introduction of residual learning alleviated the vanishing gradient problem [24]. The author of very deep convolutional networks super-resolution (VDSR) [14] applied a residual network at image super-resolution. The low-resolution image carries low-frequency information that is similar to the low-frequency information in the high-resolution image. As a result, the network only needed to learn the residual high-frequency difference between a high-resolution image and a low-resolution image. This method increased the receptive field of the network, improved its performance, and simplified the network training. In 2017, the emergence of Densenet [25] further increased the connectivity of network features. Every layer’s feature result became the input once more, allowing the network to learn more detailed feature information. To produce reconstructions that are more accurate than the actual data, Ledig et al. [26] suggested the super-resolution generative adversarial network (SRGAN). The generative adversarial network (GAN) [27] divided the network into two parts: a generative model and a discriminative model. The generative model was used for generating super-resolution images. The discriminative model was used for discriminating the gap between generated images and ground truth images. The network could train more thoroughly if the loss functions of different models were competing with one another. On the basis of SRGAN, Wang suggested an enhanced super-resolution generative adversarial network (ESRGAN) [28] and a real-world enhanced super-resolution generative adversarial network (Real-ESRGAN) [29]. In order to improve the visual quality and model performance, the ESRGAN introduced residual dense blocks and the Real-ESRGAN proposed a set of degradation models for the degradation process of the real-world.

### 2.2. Image Super-Resolution Based on Attention Mechanism

Attention mechanisms can be employed for a variety of deep learning models across many different domains and tasks [30]. In computer vision, attention mechanisms are designed to locate the areas of a picture that capture human attention with a greater priority. In 1998, Itti [31] introduced a technique that employs the remarkable information of various picture elements, locates the image’s attention points, and dynamically changes the image’s attention points to replicate the shifting process of human visual attention. The Spatial Transformer Network (STN) [32], developed by Google DeepMind in 2015, allows the network to preprocess images by learning the deformation characteristic of the picture using the affine transformation theory. This is a kind of attention model based on space. Hu et al. [33] proposed a novel architectural unit called Squeeze-and-Excitation (SE), which adaptively recalibrates channel-wise feature responses by explicitly modeling interdependencies between channels. This mechanism might cause the network to prioritize the most useful information in the input. Then, using the residual in residual (RIR) structure and SE architectural, a very deep residual channel attention network (RCAN) [34] was deployed. Through adaptive modification of the weight on the feature channel, they could control the influence of the channel on the network feature. In 2018, Woo [35] integrated spatial and channel attention and proposed the convolutional block attention module (CBAM). It can make space information condense into channel information and provide a more precise attention mechanism. Coordinate attention (CA) [36] obtains the horizontal and vertical feature information of each channel, encodes the spatial information using batch normalization (BN) to normalize the data in each batch, and stabilizes its distribution. It then fuses the spatial information through the channel attention mechanism to achieve a composite attention structure. This enhances the relationships between the deep features of pixels.

## 3. Methodology

In this paper, a generative adversarial network based on an attention mechanism and a residual module was proposed, which consists of a generation network and a discrimination network. 

The generation network is used to map low-resolution images to super-resolution images. The discriminator network is used to examine the difference between the generated super-resolution image and the original image, and the discrimination loss is added into the training of the generator network, enabling the network to better recover the true image features.

### 3.1. Generation Network

In the generation network, the network is divided into four parts: pixel matching, shallow feature extraction, deep feature mapping, and image mapping reconstruction. As shown in Figure 1, Ilr and Isr represent the input and output of the generation network, respectively.

A layer of PixelUnshuffle is included in the pixel matching module to help with pixel separation. This layer realizes down-sampling by changing the four-dimensional tensor of size (B,C,H,W) to $\left(B,C\times r^{2},H/r,W/r\right)$. By adjusting the parameter *r*, it allows the training process for 1×, 2×, and 4× super-resolution tasks to share the same network. The procedure for pixel matching is depicted as
(1)$$T_{input}=H_{pus}(I_{LR})$$
where $H_{pus}$ stands for the PixelUnshuffle layer and $T_{input}$ for the tensor output. The 4× super-resolution task networks are used as the fundamental network to share a group of networks. In the 2× super-resolution task, PixelUnshuffle splits the pixels, reduces the image size by 2 times, and increases the number of channels to 4 times. Similarly, for the 1× super-resolution task, the image size is reduced by 4 times, and the number of channels is increased to 16 times. Finally, the reconstruction process realizes the inverse process of PixelUnshuffle.

For shallow feature extraction modules, single-layer convolution is used for simple linear mapping. The shallow feature extraction process is expressed as
(2)$$F_{0}=H_{sf}(Input)$$
where $H_{sf}$ represents the mapping process with 3 × 3 convolution. 

The EARDB (enhanced attention residual dense block) structure block is used as the basic skeleton in the deep feature mapping module. The residual structure between EARDBs is shown in Figure 2, which can be expressed as
(3)$$F_{n}=H_{EARDBn}^{n}(F_{n-1})+F_{n-1}=H_{EARDBn}^{n}(H_{EARDBn-1}^{n-1}(\ldots F_{0}\ldots ))+F_{n-1}$$
where $H_{EARDBn-1}^{n-1}$ and $H_{EARDBn}^{n}$ represent the (n − 1)th and nth EARDB feature extraction structures, respectively. Three attention residual dense block (ARB) structures are connected by residual structures inside the EARDB module.

In order to achieve multiscale feature fusion and reduce network parameters, the dense structure inside the ARB is used to execute feature fusion. The dense process in ARB can be expressed as
(4)$$ARB_{output}=Conv5(Cat(X_{1},Conv4_{output},\ldots ,Conv1_{output}))$$
where Conv5 is the last convolution in the ARB block. Cat is a contact structure that combines 32-dimension growth channel output from each convolution in the ARB.

For the image mapping reconstruction process, we use two up-sampling modules to interpolate the extracted features and make the feature pixels increase 4 times. The upscaling module is made up of the nearest neighbor (NN) layer. Then, the image pixel is combined using two convolution layers. It is shown as follows:(5)$$I_{SR}=Conv_{3\times 3}^{2}(H_{up}(H_{rec}(I_{LR})+I_{LR}))$$
where $H_{rec}$ represents the reconstruction module and $H_{up}$ represents the interpolation operation.

### 3.2. Enhanced Attention

The purpose of EA is to enhance the ability of the network to find key features. The input is $T_{B\times C\times X\times Y}$, a four-dimensional tensor. B represents the number of images input into one iteration of the network batch and C represents the number of characteristic channels of the image in the network. X and Y represent the size of the channels in the X and Y directions, respectively.

The structure of RCAN [34] proves that global mean pooling can build the dependency between channels, increase the sensitivity information of the model to channels, and affect the channel weights in the image reconstruction process. Additionally, inspired by coordinate attention (CA) [36], we can effectively combine channel attention and spatial attention by associating image location information. Therefore, the EA mainly consists of two processes, coordinate information generation and coordinate attention embedding. EA is shown in Figure 3.

In the process of coordinate information generation, two mean pooling kernels are used in the X and Y directions to extract the features of position information. In X direction, it outputs a tensor of H × 1 dimension. The characteristics of row m in the X direction are as follows:(6)$$Z_{k,m}=\frac{1}{W}\underset{0\leq i\leq N}{\sum x_{k}(i,m)}$$
where $Z_{k,m}$ represents the characteristics of the kth channel in m line, and $x_{k}$ represents the characteristics of the kth channel.

Similarly, we can achieve the characteristics of Y direction for a tensor of 1 × *W*. The characteristics of the nth row and the kth channel in Y direction are shown as follows:(7)$$Z_{k,n}=\frac{1}{W}\underset{0\leq j\leq M}{\sum x_{k}(j,n)}$$

Finally, M × N dimension tensor is compressed into two low dimensional tensors M × 1 and 1 × N. 

To preserve the key points in the channel, we use two maximum poolings to record the maximum values of the rows and columns. The X-direction feature tensor of dimensions H × 1 and the Y-direction feature of dimensions 1 × W are finally obtained. The pooling process in X direction is shown as
(8)$$E_{k,m}=\underset{0\leq i\leq N}{Max}(x_{k}(i,m))$$

The pooling process in Y direction is expressed as
(9)$$E_{k,m}=\underset{0\leq j\leq M}{Max}(x_{k}(j,n))$$

The result in Equations (6)–(9) transform into a tensor in (m + n) × 2 through dimension change and contact operation.

For the coordinate attention-embedding process, it needs to encode all the position features and generate attention parameters. These parameters serve to emphasize the area of interest within the picture. In addition, the coding process should also consider the relationship between channels based on location information.

Firstly, coding map characteristics are obtained by 1 × 1 convolution and non-linear mapping module. In CA, the author adds BN to facilitate network training. However, it has been confirmed in several models such as ESRGAN [28] that BN leads to the loss of image information and the smoothing of strong changes between pixels in the super-resolution task. It is not conducive to the reconstruction of image details. After coding and non-linear mapping, we decode the feature into two tensors Tx and Ty of X and Y dimensions. Then, the two tensors are transformed by convolution, respectively. The decoding process is shown as
(10)$$g_{x}=\sigma (Conv_{1\times 1}(T_{x}))$$
and
(11)$$g_{y}=\sigma (Conv_{1\times 1}(T_{y}))$$
where $g_{x},g_{y}$ are the attention features of x direction and y direction obtained by the EA. Here, $Conv_{1\times 1}$ is the convolutional decoding process. $T_{x},T_{y}$ are the tensors after encoding and non-linear mapping. Finally, the input feature is multiplied by the attention feature result, which is output for EA.

### 3.3. Discriminator and Loss Function

The discrimination network is shown in Figure 4. Input is the super-resolution image generated by the generation network, and output is the probability that the super-resolution image is close to the real image. The network structure mainly refers to the design idea of the VGGnet [37], which consists of convolution, preLU, and BN. It contains 8 convolutional layers; the convolution kernel size is 3 × 3, and the convolution dimension gradually increases from 64 dimensions to 512 dimensions. After obtaining the deep features in the convolutional layer, the final probability value is obtained by two fully connected layers, one preLU layer, and one sigmod layer.

The generation network loss function is denoted by $L_{G}$ and is shown as follows: it includes content loss function, the perceptual loss function and the adversarial loss function.
(12)$$L_{G}=L_{per}+\lambda_{1}L_{L1}+\lambda_{2}L_{G}^{Ra}$$

In Equation (12), $\lambda_{1}$ and $\lambda_{2}$ are the weighting coefficients used to balance the two loss functions. 

The content loss function is used to evaluate the L1 distance of the image Isr generated by the generation network from the original image Ihr. 

The perceptual loss $L_{per}$ is defined by a pretrained VGG16 network. The perceptual loss function is defined as the Euclidean distance between the features of the reconstructed image Isr and the real image IHR. It is expressed as
(13)$$L_{per}=E\{||\varphi [G(x)]-\varphi (y)|{|}_{2}^{2}\}$$
where φ[G(x)] represents the feature map of the generated super-resolution image through vgg16, and B is the feature map of the original high-resolution image through vgg16.

The adversarial loss $L_{G}^{Ra}$ is used to judge the image generated by the network, and the adversarial loss function of the discriminator seeks to maximize the proportion of accurate evaluations. The loss function is expressed as
(14)$$L_{D}=-E\{\log D_{Ra}[y,G(x)]\}-E\{\log\{1-D_{Ra}[G(x),y]\}\}$$
where $D_{Ra}$ represents the output of adversarial network.

The purpose of the generator’s adversarial loss function is to minimize the probability of the correct judgment, which is expressed as
(15)$$L_{D}=-E\{\log\{1-D_{Ra}[y,G(x)]\}-E\{\log\{D_{Ra}[G(x),y]\}\}$$
where X is the original image and G(x) is the image generated by the generated network.

## 4. Experiments

### 4.1. Discriminator and Loss Function

In order to compare with other SR algorithms, we use the most common training datasets DIV2K [38] and Flicker2K [39] for the training dataset DF2K. Among them, DIV2K includes 1000 2K resolution images, and Flicker2K includes 1450 2K resolution images. These images were cropped into 48,115 pieces of 400 × 400 pixel images, and the low-resolution images are obtained via the bicubic down-sampling.

In the training process, Set5 [40] is adopted for validation after every 500 iterators. Additionally, for the final result verification process, public benchmark datasets Set5 and Set14 [41] are employed to evaluate our proposed network.

During the imaging process, the scattering and refraction of electromagnetic waves will produce periodic stripes in addition to the noise. In the Fourier frequency spectrum, these periodic stripes have characteristic frequency points with high amplitude [42]. In order to apply the network to super-resolution reconstruction of terahertz images, we design a terahertz image degradation model to simulate real terahertz images. The degenerate expression is as follows:(16)$$o_{1}(x,y)=IFFT\{FFT[i(x,y)*PSF(x,y)]*Mask\}$$

$O_{1}\left(x,y\right)$ is a simulated terahertz image that has been degraded, while i(x,y) is the original image. Firstly, the picture is blurred through PSF(x,y), which is a Gaussian blur kernel. Afterwards, we use the fast Fourier transform (FFT) to convert the image to the frequency domain and multiply it with a multiplicative *Mask*. *Mask* is a matrix used to increase the amplitude value of spectral feature points. The position of the *Mask* is the characteristic frequency points with high amplitude positions mentioned above and it is usually 1/4 height up and down from the vertical position of the image center. Finally, degraded terahertz-simulated images are obtained by the inverse Fourier transform (IFFT).

In addition, we build a dataset of tomography results, and apply the degradation algorithm to this dataset. It includes 352 pictures, 340 for network training and 12 for testing and verification.

Based on the original image $I_{HR}$ and the reconstructed image $I_{SR}$, the peak signal-to-noise ratio (PSNR) and the structural similarity index (SSIM) are calculated to evaluate the network effect. The PSNR is expressed as
(17)$$PSNR=10\times \mathrm{lg}(\frac{255^{2}}{MSE})$$
where
(18)$$MSE=\frac{1}{M\times N}[\sum_{i=1}^{M}\sum_{j=1}^{N}{\left(f(i,j)-\hat{f}(i,j)\right)}^{2}]$$M and N are the height and width of an image, respectively. f(i,j) represents the grayscale values of all pixels in the original image and $\hat{f}\left(i,j\right)$ represents the grayscale values of all pixels in the reconstructed image. The reconstructed image looks most like the original image when the PSNR value is high.

The SSIM is formulated as
(19)$$SSIM(F,f)=\frac{\left(2\mu_{F}\mu_{f}+c_{1})(2\sigma_{Ff}+c_{2}\right)}{\left(\mu_{F}{}^{2}+\mu_{f}{}^{2}+c_{1})(\sigma_{F}{}^{2}+\sigma_{f}{}^{2}+c_{2}\right)}$$

F is the original reference image, f is the image to be evaluated, μ is the image gray level mean, and σ is the image gray level variance. C1 = k1 × L and C2 = k2 × L. L is the image gray level, where L is the image gray level and k1 and k2 are equal to 0.01 and 0.03, respectively. SSIM obtains quantitative values by comparing the luminance, contrast, and structure of the original image and the reconstructed image. The larger the SSIM value, the closer the reconstructed image is to the original image.

Texture features in pictures with a high PSNR or SSIM may not match to the visual habits of the human eye. NIQE is a non-parametric evaluation index that measures the impact of image super-resolution by comparing the Gaussian distributions of the original picture and the super-resolution image. NIQE evaluates image quality by equation as
(20)$$D(v_{1},v_{2},\sum_{1},\sum_{2})=\sqrt{\left({\left(v_{1}-v_{2}\right)}^{T}{\left(\frac{\sum_{1}+\sum_{2}}{2}\right)}^{-1}(v_{1}-v_{2})\right)}$$
$v_{1},\sum_{1}$ is the mean and variance of the original image Gaussian distribution, and $v_{2},\sum_{2}$ is the mean and variance of the reconstructed image Gaussian distribution.

### 4.2. Training Details

Our model is trained using the PyTorch framework with an NVIDIA RTX 1660Ti GPU. In the pretraining, the L1 loss function is used to train the generation network and a model with a high PSNR is obtained. The optimizer is set to Adam optimizer, and the initial learning rate is 2 × 10^−4^. The optimizer parameters are β1 = 0.9, parameter β2 = 0.99, and batch size = 16.

In training process, we employ two sets of learning rate adjustment strategies. Firstly, we employ the multistep learning rate (MultiStepLR), a technique that gradually decreases the learning rate. It can reduce the learning rate by fifty percent for every 25,000 iterations.

Secondly, the cosine annealing learning rate algorithm (CosineAnnealingLR) is used to adjust the learning rate. The characteristic of this algorithm is that the learning rate initializes at a small value, and then the rate can rise when the model becomes stable. After that, the learning rate declines gradually. In addition, the training process includes multiple CosineAnnealingLR cycles, and the learning rate of each cycle is initialized.

In the experiment, we set the CosineAnnealingLR algorithm’s learning cycle to 30,000, 30,000, and 40,000 iterations. Additionally, the learning rate for the MultiStepLR method is modified every 25,000 iterations. Both of them have a total training time of 10,000 iterations.

The training curves of the two algorithms in training 9block-x4-EARDB are depicted in Figure 5, with both algorithms using 100,000 iterations to obtain the final model. It can be observed that the PSNR experiences a sudden drop after 30,000 and 60,000 iterations when using the CosineAnnealingLR algorithm. However, the curve rapidly rises again after the learning rate restarts. Compared to the MultiStepLR algorithm, the CosineAnnealingLR algorithm results in a 0.13 dB improvement in performance.

After the pretraining, this model is used as the initial model of the generated network and trained with the discriminant network. The initial learning rate is set to 10^−4^; Adam optimizer and MultiStepLR algorithm are used to train the generative adversarial network.

### 4.3. Ablation Study

Under the same training settings, to demonstrate the effectiveness of our proposed architecture, we test different attention networks on the original network. The original network structure is shown in Figure 1.

The quantitative comparisons of different attention networks for ×2 SR task and ×4 SR task are depicted in Table 1 on the datasets Set5 and Set14.

In order to compare the CA mechanism with other advanced attention mechanisms, we have added different attention mechanisms to the same network structure, the residual dense block (RDB) network. In Table 1, the bicubic adopts the linear interpolation method. RDB is a network without the attention structure, and the rest of the networks add SE [33], CBAM [35], CA [36], and EA, respectively. In addition, the result of the EARDB network is used as a pretraining model to train generative adversarial network with enhanced attention residual dense block (EARDB-GAN).

The attention structure is added after the first four convolutions in the block. The number of blocks is set to 9 and the number of iterations to 100k iterators.

Through comparison, we find that EA has the best effect on ×2 task and ×4 task. On ×4 task, PSNR and SSIM have a greater improvement effect, which indicates that the more pixels the more obvious the EA attention mechanism is on the feature. We also compared the results of the generated network and the discriminant network after each iteration, and found that PSNR and SSIM decreased to some extent, but NIQE reached the maximum.

The reconstruction results of EARDB, EARDB-GAN network, and bicubic algorithm with ×4 image super-resolution are shown in Figure 6. It can be found that through our proposed network, the image details have been better reconstructed. Despite the fact that the EARDB network can achieve a higher PSNR, the images obtained by EARDB-GAN are more similar to real images.

In order to prove the improvement effect of the EARDB-GAN, we compare it with SRGAN [26] and ESRGAN [28] under the same training conditions. The results in Table 2 show that our network is a lightweight model, which reduces the parameters by one time and achieves the same effect as ESRGAN.

In this section, we test different generation networks in Set14 with varying numbers of blocks. The experimental results with a scaling factor of ×2 in five models are shown in Table 3. It can be seen that the network has the best effect in EARDBx4 with 9 blocks, where PSNR and SSIM achieve the biggest value. In a deeper situation, the model continues to increase the number of network layers, which does not significantly improve the objective evaluation indicators, but increases the number of model parameters. 

In order to verify the effect of the algorithm on terahertz images, we use the terahertz image database to resume training with the network EARDB-GAN, which is trained by DF2K. Additionally, the terahertz image database contains 352 computer-generated geometric images. The dataset includes images of various combinations of geometric shapes, such as triangles, circles, pentagons, etc. Figure 7 shows some images of this database, degraded images, and images after EARDB-GAN network reconstruction. It can be seen that the degradation algorithm has successfully simulated some problems of terahertz images, such as their low resolution, blurry edges, and fringe artifacts. Using such a dataset for training, the network can learn the degradation process of terahertz images. From the reconstructed images, the network in this paper can accurately restore the terahertz image details and retain the object contour.

Figure 8 shows a group of real terahertz images. These images were preprocessed using the wavelet adaptive threshold denoising algorithm [43], which employs wavelet decomposition and adaptively adjusts the denoising threshold and wavelet reconstruction to obtain images that are denoising with smoother edges. Finally, these preprocessing images are reconstructed by EARDB-GAN. It can be found that the method in this paper has a good effect on the super-resolution task of terahertz images.

## 5. Conclusions

In this paper, we propose a super-resolution reconstruction method for terahertz images based on a residual generative adaptive network with an enhanced attention mechanism. The network’s key parameters can be adaptively updated using the attention module, and pixel coordinate information can be incorporated into the attention mechanism. Efficient residual dense connection blocks are used to realize the multiscale information fusion of the image. Extensive quantitative and qualitative experiments demonstrate that our method outperforms most state-of-the-art attention mechanisms. 

The network’s training effect has been improved through the periodic simulated annealing training method. 

To apply the network to terahertz image super-resolution reconstruction, a terahertz image training dataset and image degradation algorithm have been established. The experiments show that our algorithm has a significant impact on terahertz image reconstruction.

## Figures and Tables

**Figure 1 entropy-25-00440-f001:**
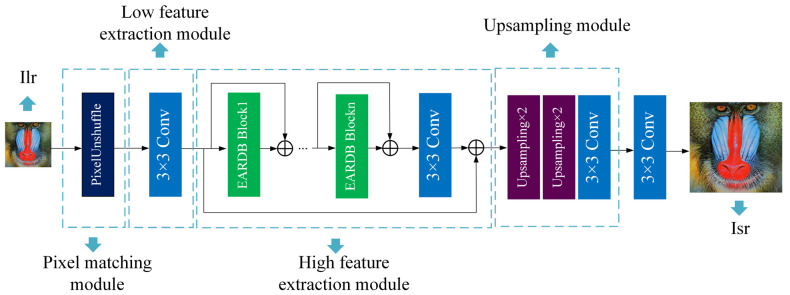
An overview of our generation network.

**Figure 2 entropy-25-00440-f002:**
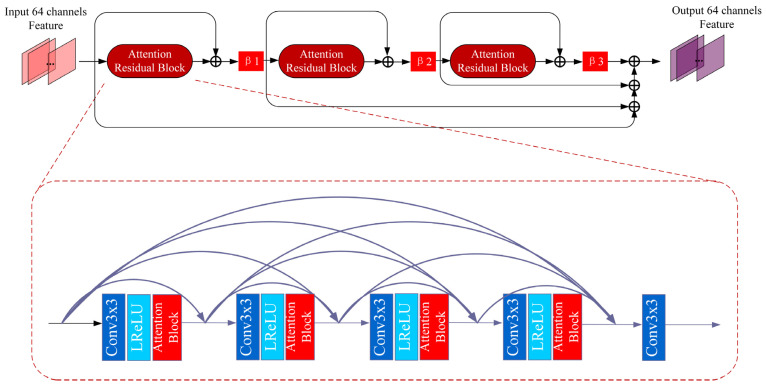
The architecture of EARDB block.

**Figure 3 entropy-25-00440-f003:**
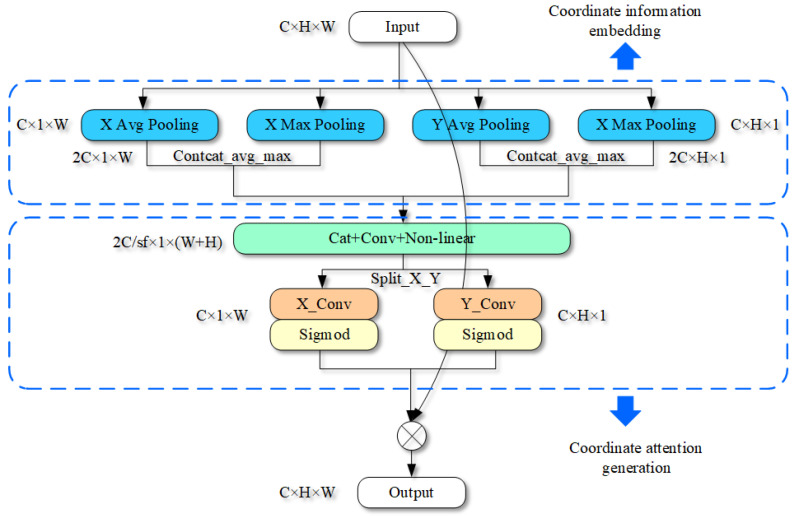
The architecture of enhanced attention.

**Figure 4 entropy-25-00440-f004:**
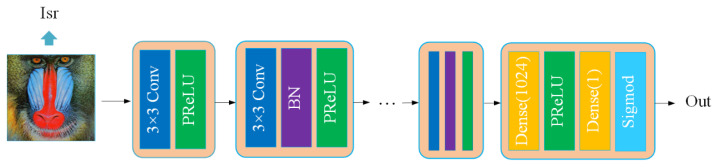
An overview of our discriminator network.

**Figure 5 entropy-25-00440-f005:**
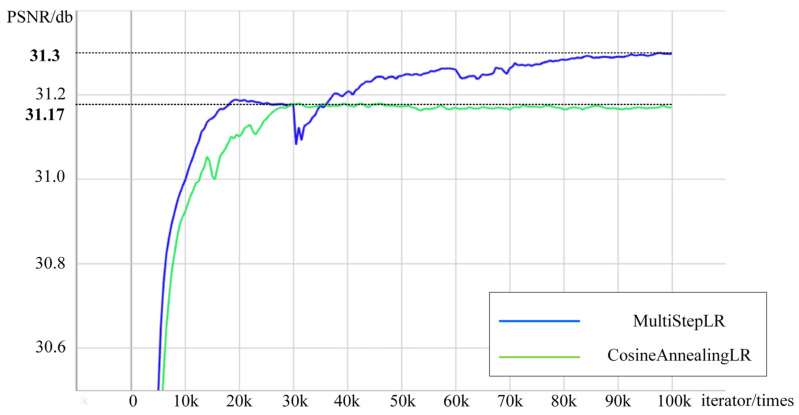
The training curves of two learning rate adjustment schemes.

**Figure 6 entropy-25-00440-f006:**
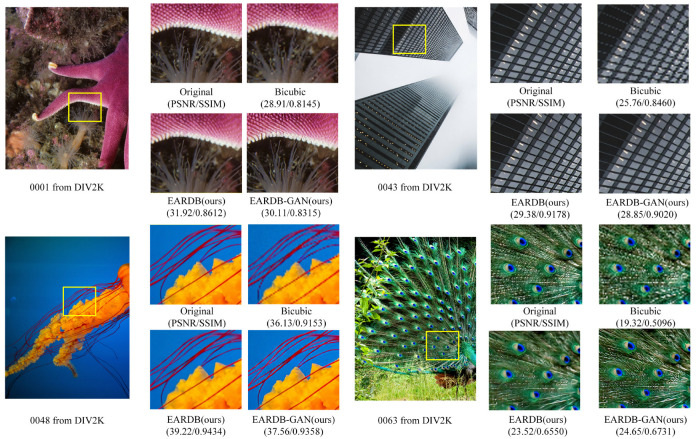
Visual comparison of ×4 super-resolution images on the DIV2K datasets.

**Figure 7 entropy-25-00440-f007:**
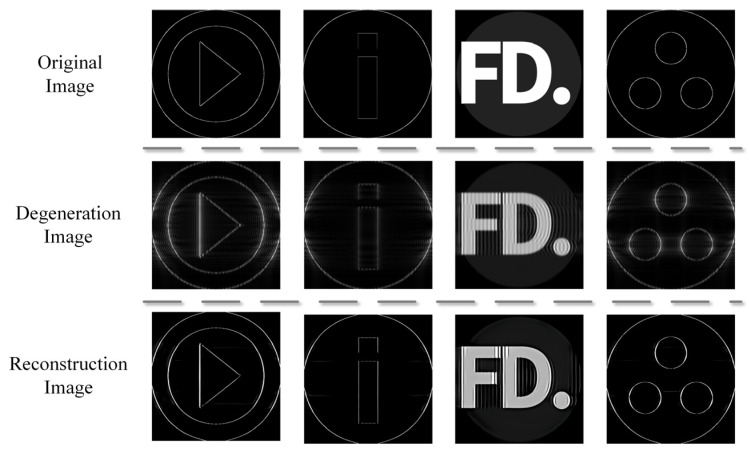
Application of degeneration algorithm on terahertz dataset and reconstruction effect of our EARDB-GAN network.

**Figure 8 entropy-25-00440-f008:**
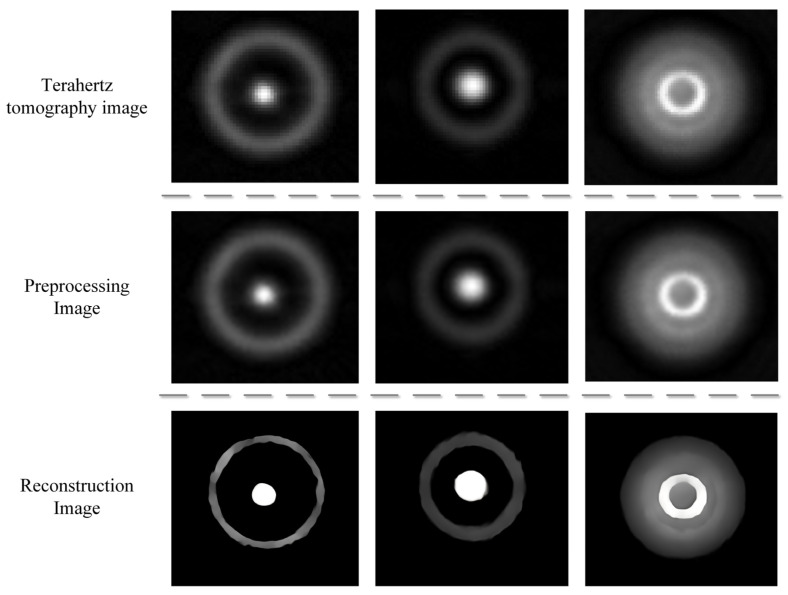
Reconstruction of real terahertz image after wavelet preprocessing.

**Table 1 entropy-25-00440-t001:** Quantitative results of several SR models with attention architecture at scaling factors of ×2 and ×4 (average PSNR/SSIM/NIQE). The best performance is highlighted in red.

Method	Scale	Params	Set5	Set14
		(K)	PSNR/SSIM/NIQE	PSNR/SSIM/NIQE
Bicubic	×2	0	33.52/0.9230/6.2358	30.21/0.8683/5.5834
RDB	×2	27,715	36.72/0.9435/6.7594	32.69/0.8988/5.8812
SERDB	×2	27,935	36.89/0.9483/6.7312	33.02/0.9075/5.8619
CBAMRDB	×2	28,150	36.93/0.9501/6.7248	33.08/0.9083/5.8405
CARDB	×2	28,375	37.01/0.9510/6.7101	33.23/0.9108/5.8322
EARDB	×2	30,470	37.14/0.9527/6.7262	33.45/0.9113/5.8326
EARDB-GAN	×2	30,470	36.95/0.9491/6.2162	33.17/0.9091/5.5138
Bicubic	×4	0	28.41/0.8091/7.2812	25.97/0.7023/6.4523
RDB	×4	27,695	30.61/0.8813/7.4811	27.53/0.7729/6.6979
SERDB	×4	27,915	30.85/0.8852/7.4631	27.75/0.7782/6.6810
CBAMRDB	×4	28,130	30.92/0.8891/7.4566	27.91/0.7829/6.6731
CARDB	×4	28,355	31.11/0.8908/7.4392	28.03/0.7853/6.6607
EARDB	×4	30,450	31.30/0.8913/7.4401	28.20/0.7890/6.6725
EARDB-GAN	×4	30,450	31.03/0.8901/7.2293	27.86/0.7851/6.4281

**Table 2 entropy-25-00440-t002:** Several SR models with GAN at scaling factors of 2 and 4 yielded quantitative results (average PSNR, SSIM, and NIQE). The best performance is highlighted in red.

Method	Scale	Params	Set5	Set14
		(K)	PSNR/SSIM/NIQE	PSNR/SSIM/NIQE
SRGAN	×2	110,870	36.83/0.9428/6.2257	32.81/0.9041/5.5174
ESRGAN	×2	130,950	36.91/0.9483/6.2203	32.95/0.9067/5.5153
EARDB-GAN	×2	30,470	36.95/0.9491/6.2162	33.17/0.9091/5.5138
SRGAN	×4	110,830	30.95/0.8879/7.2317	27.79/0.7831/6.4297
ESRGAN	×4	130,910	31.01/0.8892/7.2303	27.86/0.7845/6.4282
EARDB-GAN	×4	30,450	31.03/0.8901/7.2293	27.86/0.7851/6.4281

**Table 3 entropy-25-00440-t003:** Comparison network with the number of RDB. All the other settings are strictly the same. The best performance is highlighted in red.

Network	EARDBx4 7 Blocks	EARDBx4 8 Blocks	EARDBx4 9 Blocks	EARDBx4 10 Blocks	EARDBx4 11 Blocks
Params	23,752 K	27,101 K	30,450 K	33,799 K	37,148 K
PSNR/SSIM	28.13/0.7881	28.17/0.7887	28.20/0.7890	28.19/0.7890	28.14/0.7883

## Data Availability

Not applicable.

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
