# Peer review of "Super-Resolution Reconstruction of Terahertz Images Based on Residual Generative Adversarial Network with Enhanced Attention"

_entropy, 2023, doi:10.3390/e25030440_

Round 1

Reviewer 1 Report

The authors propose a generative adversarial network (GAN) super resolution (SR) in Terahertz imaging. It uses "enhanced attention" (EA), which combines existing ideas of channel and coordinate attention in a residual network architecture. Generator losses consist of L1 distance and L2 VGG-feature distances. In combination with the discriminator, a straightforward adversarial loss is used. The authors claim that this network offers improved SR in general and works particularly well in Terahertz tomography. They also mention in the introduction that they contribute a new Terahertz database.

Novelty

From the relatively coarse descriptions of the networks, the approach itself seems technically sound, which is confirmed by the experiments. The individual components of the network are inspired by previous approaches with some alterations and in general a relatively lightweight network. Together with the relatively rare application of deep learning in Terahertz imaging so far, the amount of novelty could be sufficient, but should be better differentiated from previous approaches. There is some discussion of related work in Section 2, but in the methodology section, no citations can be found that mark prior work. For a reader who is not deeply familiar with other SR and attention methods, it is hard to spot where old components end and new variations begin. Even at points where the authors try to put their work into context (references to RCAN and CA), the reader might have to jump back to Section 2.

Related Work

In general, the discussion of related work could be improved throughout the whole manuscript. It is good, that the authors discuss Terahertz imaging in general in the very beginning, but they only give a few concrete applications as examples. Citing one or multiple review papers might complement this nicely, e.g. for Terahertz imaging in general Christian et al. (2010) or Valušis et al. (2021), or more concretely Guillet et al. (2014) for Terahertz tomography.

On the deep learning side, the authors cite several papers on super resolution as well as on attention, but both are obviously broad fields, for which it would be best to cite survey articles like Wang et al. (2020) and Brauwers/Fraincar (2021). Citing some recent developments would also be a good addition, for instance SR with probabilistic diffusion (Rombach et al. 2022). On the SR-side the authors also must argue convincingly why their comparisons in the experimental section are adequate. They compare against several different attention mechanisms in an ablation study and against SRGAN/ESRGAN. But why among the huge body of work existing on SR are those the adequate reference points? For instance, they mention RCAN prominently, which yields good quality in survey papers, but compare against SRGAN/ESRGAN instead.

Also, there are more publications that combine Terahertz imaging with deep learning and should be cited, for instance Fan et al. (2021), Su et al. (2022), and Yang (2022). Again, in the super-resolution case it must be justified why there is no direct comparison (e.g. because the imaging application is too different) or experiments should be added.

Moreover, there are some very obvious gaps in the citations that must be rectified:
- When using GANs, it is appropriate to cite Goodfellow et al. (2020).
- Similarly, residual architectures go back to He et al. (2016).
- ESRGAN is compared to, but not cited (Wang et al. 2018).
- SE is mentioned and considered in ablation, but not cited (Hu et al. 2018)

Experiments:

The overall experimental setup is good in that it uses public datasets DIV2K and Flicker2K for training, as well as public benchmarks [29]. Also, synthetic simulation of Terahertz images is a valid strategy and the training process is reasonably well documented for reproducibility.

The layout of the experimental section with short text blocks interleaved by images and tables seems quite confusing at first. As soon as it becomes clear that text blocks always reference the directly adjacent figures, even if those are not explicitly referenced, the experimental section becomes a bit more accessible. A cleaner layout (top aligned tables and figures which are properly referenced in the text) would make it easier to navigate, though. Tables and Figures are not always well-referenced in the text and it is quite cumbersome to follow the experiments. For instance, in line 359, Table 3 is mislabelled as Table 1, Table 1 on p. 10 spreads over multiple pages, and Table 2 is not references in 350-353.

Additional confusion arises from the many acronyms, which are not always well-documented. Previous works like SRGAN and ESRGAN are only marked with citations in the introduction, but not hear (ESRGAN is not properly cited at all). Other differences are also hard to find in the paper, e.g. what is EAGAN? How is it different from EARDB-GAN and how does that differ from EARDB. (The latter is at least somewhat easy to guess.)

It is also unclear if the database of Terahertz images will be published? There are also little details on where these images come from, what content they contain, with what devices they were obtained etc.

Results of the SR comparison are encouraging and are okay for Terahertz images. Since there are only little true Terahertz experiments, it is unclear how much of an improvement it really is compared to existing methods for Terahertz tomography. In particular, there is no comparison to other methods designed for that setting. Also, the method produces clear directional biases as seen in Fig. 8. The input shapes are almost rotationally invariant, but the sharp reconstructions are not. Thus, small deviations in the blurry reconstructions seem to result in larger deviations in the sharp SR reconstruction.

Presentation:

The presentation could often be further improved by leading with an explanation of the intuitive idea. For instance, the authors first introduce the term "PixelUnshuffle" layer (which will be unfamiliar to readers who have not used it in PyTorch). They first use this term, then define it by the quite superficial formula (1) and explain its relevancy for the task at hand in the following paragraph. It would be much more efficient to describe the problem that is solved by unshuffling *first*, then providing a cleaner mathematical formulation. For the latter, at the very least input and output dimensions should be specified.

Acronyms are often unclear:
- Fig. 2 refers to ARDB blocks, presumably that's the same as ARB?
- RCAN refers to Zhang et al. [19], not referenced again in 3.2, same applies to CA [25].
- bn used for batch normalisation in 215 without explanation
- 216: ESRGAN and claim on bn without citation (Wang et al. 2018)

The manuscript should be checked again for formal errors, here just a few examples:

There are formal references in the errors, for instance:
- There are many strange journal abbreviations, e.g. J.o.b. for J Biophotonics in [1]. Strangely, proceeding titles are not abbreviated.
- Terahertz is sometimes uppercase and sometimes lowercase in titles
- Obvious errors like "Proceedings of the Proceedings" in [12].
There are probably more than these few samples.

l. 58: It is not really clear what "ignoring the reconstruction of visual features" is referring to, here.

Uppercase for "Terahertz" is inconsistent, sometimes the authors use uppercase, sometimes lowercase.

Sometimes singular terms are used without an article that either should be converted to plural or  complemented with an article. Some examples:
l. 11 ... waves are widely used ...

In other case, a definite article is used when a general class of algorithms, where it should rather be removed and replaced by plural - or be replaced by an indefinite article. One example:
l. 51: "based on convolutional neural networks" or "based on a convolutional neural network".

l. 65 Zhang et al missing "."
l. 144 grammar issues: to aid -> aids?

Recommendation:

In principle, deep learning SR for Terahertz imaging is a viable application which received relatively little attention so far. The approach itself does not seem revolutionary, but reasonable.

The manuscript however requires a rework w.r.t. multiple aspects, including a more appropriate literature discussion, more clear differentiation of the novelties to existing work, and a more convincing presentation of results.

Therefore, I recommend a major revision.

References:

Brauwers, Gianni, and Flavius Frasincar. "A general survey on attention mechanisms in deep learning." IEEE Transactions on Knowledge and Data Engineering (2021).

Fan, Lei, et al. "Fast and High-Quality 3-D Terahertz Super-Resolution Imaging Using Lightweight SR-CNN." Remote Sensing 13.19 (2021): 3800.

Goodfellow, Ian, et al. "Generative adversarial networks." Communications of the ACM 63.11 (2020): 139-144.

He, Kaiming, et al. "Deep residual learning for image recognition." Proceedings of the IEEE conference on computer vision and pattern recognition. 2016.

Hu, Jie, Li Shen, and Gang Sun. "Squeeze-and-excitation networks." Proceedings of the IEEE Conference on Computer Vision and Pattern Recognition. 2018.

Guillet, Jean Paul, et al. "Review of Terahertz tomography techniques." Journal of Infrared, Millimeter, and Terahertz Waves 35.4 (2014): 382-411.

Jansen, Christian, et al. "Terahertz imaging: applications and perspectives." Applied optics 49.19 (2010): E48-E57.

Rombach, Robin, et al. "High-resolution image synthesis with latent diffusion models." Proceedings of the IEEE/CVF Conference on Computer Vision and Pattern Recognition. 2022.

Su, Wen-Tai, et al. "Seeing Through a Black Box: Toward High-Quality Terahertz Imaging via Subspace-and-Attention Guided Restoration." European Conference on Computer Vision. Springer, Cham, 2022.

Valušis, Gintaras, et al. "Roadmap of Terahertz imaging 2021." Sensors 21.12 (2021): 4092.

Wang, Zhihao, Jian Chen, and Steven CH Hoi. "Deep learning for image super-resolution: A survey." IEEE transactions on pattern analysis and machine intelligence 43.10 (2020): 3365-3387.

Wang, Xintao, et al. "ESRGAN: Enhanced super-resolution generative adversarial networks." Proceedings of the European conference on computer vision (ECCV) workshops. 2018.

Yang, Xiuwei, et al. "Super-resolution reconstruction of terahertz images based on a deep-learning network with a residual channel attention mechanism."
Applied Optics 61.12 (2022): 3363-3370.

Author Response

Please see the attachment. We have packaged our response to your comments and the revised paper in a PDF file.

Reviewer 2 Report

This paper proposes a super resolution approach for terahertz images based on a residual generative adaptive network in combination of an enhancement mechanism. My opinion is that there's material to publish but my opinion is just at first glance: The paper needs a strong improvement.

Specifically:

- English needs a strong revision. Some sentences are hard to follow

- Laurent Ltty is Laurent Itty: this is just one of the examples that show that the paper needs a careful revisitation

- In Introduction you claim:

3. We use the cosine annealing algorithm to improve the network training process, speed up the training process, and effectively improve the network performance.

It means that there's a speed up: which is the original training time and which is the arrival one?

- Please, may you give more details and comments about epochs, eventual overfitting etc. that may help the reader to better understand the network behavior?

- Are you sure that SNR and SSIM are able to really catch texture recover? Usually, a final check is always visual. May you insert another example that shows better this aspect --- that is fundamental in this paper?

- I saw that in one of the experiments there's a wavelet preprocessing. Ther's no explaination about its presence and its effect: can you spend more words about it?

- I have some doubts about the degradation model. It seems to me mainly linear.

In this case the network usually gives good results. Have you tried to add

some noise (or something similar) to break the linearity in the degradation? May you add some examples?

By the way: may define exactly which is 'mask'?

I think that the paper 'seems' interesting but has a lot of work to do. I want to help you with a major revision though I guess the work to accomplish the comments  above could require a bit of time.

Author Response

(The authors gave the same response as above.)

Round 2

Reviewer 1 Report

The authors have addressed all major scienfitic points raised in the first review round. I recommend doing a final formal polishing pass checking for smaller grammar issues and typos before publication but have no additional requests for mandatory changes.

Reviewer 2 Report

Authors have implemented the required changes. The paper can be accepted in this form.